Chemotherapeutic resistant cholangiocarcinoma displayed distinct intratumoral microbial composition and metabolic profiles

Sitthirak Sirinya 1 2
Suksawat Manida 1 2 3
Phetcharaburanin Jutarop 1 2 3
Wangwiwatsin Arporn 1 2 3
Klanrit Poramate 1 2 3
Namwat Nisana 1 2 3
Khuntikeo Narong 1 3 4
Titapun Attapol 1 3 4
Jarearnrat Apiwat 1 4
Sangkhamanon Sakkarn 1 5
Loilome Watcharin watclo@kku.ac.th 1 2 3
1 Cholangiocarcinoma Research Institute, Khon Kaen University , Khon Kaen , Thailand
2 Department of Biochemistry, Faculty of Medicine, Khon Kaen University , Khon Kaen , Thailand
3 Khon Kaen University International Phenome Laboratory, Khon Kaen University , Khon Kaen , Thailand
4 Department of Surgery, Faculty of Medicine, Khon Kaen University , Khon Kaen , Thailand
5 Department of Pathology, Faculty of Medicine, Khon Kaen University , Khon Kaen , Thailand
Young Howard
Electronic publication date: 2022 Aug 16
Publication date: 2022
Volume: 10
Electronic Location ID: e13876
Received 2022 Apr 27; Accepted 2022 Jul 19
Copyright: ©2022 Sitthirak et al.
Copyright year: 2022
Copyright holder: Sitthirak et al.
License: This is an open access article distributed under the terms of the Creative Commons Attribution License, which permits unrestricted use, distribution, reproduction and adaptation in any medium and for any purpose provided that it is properly attributed. For attribution, the original author(s), title, publication source (PeerJ) and either DOI or URL of the article must be cited.
License URL: https://creativecommons.org/licenses/by/4.0/

Keywords: Cholangiocarcinoma, Microbiome, Metabolome, Intratumoral Bacteria, Chemotherapeutic resistant

Funding: The National Research Council of Thailand through Fluke Free Thailand Project The NSRF under the Basic Research Fund of Khon Kaen University through Cholangiocarcinoma Research Institute to Watcharin Loilome Invitation Research Grant allocated to Sirinya Sitthirak IN64123 The Graduate school of Khon Kaen University 621JH102 This study was supported by a grant of the National Research Council of Thailand through Fluke Free Thailand Project and the NSRF under the Basic Research Fund of Khon Kaen University through Cholangiocarcinoma Research Institute to Watcharin Loilome and a grant from the Invitation Research Grant (IN64123) allocated to Sirinya Sitthirak. Sirinya Sitthirak was awarded a scholarship from the Graduate school of Khon Kaen University (Grant No. 621JH102). The funders had no role in study design, data collection and analysis, decision to publish, or preparation of the manuscript.

==============================
Background

Cholangiocarcinoma (CCA) is a malignancy of the cholangiocytes. One of the major issues regarding treatment for CCA patients is the development of chemotherapeutic resistance. Recently, the association of intratumoral bacteria with chemotherapeutic response has been reported in many cancer types.

Method

In the present study, we aimed to investigate the association between the intratumoral microbiome and its function on gemcitabine and cisplatin response in CCA tissues using 16S rRNA sequencing and 1H NMR spectroscopic analysis.

Result

The results of 16S rRNA sequencing demonstrated that Gammaproteobacteria were significantly higher in both gemcitabine- and cisplatin-resistance groups compared to sensitive groups. In addition, intratumoral microbial diversity and abundance were significantly different compared between gemcitabine-resistant and sensitive groups. Furthermore, the metabolic phenotype of the low dose gemcitabine-resistant group significantly differed from that of low dose gemcitabine-sensitive group. Increased levels of acetylcholine, adenine, carnitine and inosine were observed in the low dose gemcitabine-resistant group, while the levels of acetylcholine, alpha-D-glucose and carnitine increased in the low dose cisplatin-resistant group. We further performed the intergrative microbiome-metabolome analysis and revealed a correlation between the intratumoral bacterial and metabolic profiles which reflect the chemotherapeutics resistance pattern in CCA patients.

Conclusion

Our results demonstrated insights into the disruption of the microbiome and metabolome in the progression of chemotherapeutic resistance. The altered microbiome-metabolome fingerprints could be used as predictive markers for drug responses potentially resulting in the development of an appropriate chemotherapeutic drug treatment plan for individual CCA patients.

Introduction

Cholangiocarcinoma (CCA) is a malignancy of the bile duct epithelia or cholangiocytes with its highest incidence in Thailand, especially in the northeastern region (Alsaleh et al., 2019). This region has high incidence of liver fluke (Opisthorchis viverrini (Ov)) infection which is recognized as the major risk factor of cholangiocarcinoma development (Piratae et al., 2012). Nowadays, surgical resection is considered the standard treatment for the patients with CCA. However, surgical treatment still provides a low survival rate (Aljiffry, Walsh & Molinari, 2009), and it leads to better treatment outcomes for the CCA patients who have been diagnosed at an early stage (Khuntikeo et al., 2015). Moreover, surgical resection in combination with adjuvant chemotherapy provides a higher survival rate when compared with the surgery alone (Wirasorn et al., 2013). Common chemotherapeutic regimens used in clinical treatments for biliary tract cancer patients are gemcitabine and gemcitabine plus cisplatin (Valle et al., 2010). Okusaka et al. (2014) demonstrated that the combination of cisplatin and gemcitabine provide the best benefit in terms of extending survival for CCA patients. However, the major issue regarding chemotherapeutic drug treatment for CCA patients is the development of chemotherapeutic resistance phenotypes, especially those involving multi-drug resistance (MDR) (Chan & Coward, 2013).

In 2019, Suksawat et al. (2019, 2022) evaluated the chemotherapeutic response of CCA patients to gemcitabine and gemcitabine plus cisplatin treatments using a histoculture drug response assay (HDRA) and metabolic profiling. In their results, the TCA cycle intermediates, alpha-D-glucose and ethanol may serve as predictive biomarkers for gemcitabine and cisplatin sensitivity in the tumor tissue of CCA patients. Moreover, methyl-guanidine may be used as a serum predictive biomarker for gemcitabine sensitivity (Suksawat et al., 2022).

Evidence has shown that the gut microbiota can shape the efficiency of cancer therapy (Ma et al., 2019). Studies have also demonstrated that the alteration of microbiota composition have various effects on tumor biology, including the transformation process, tumor progression, and the response to anti-cancer therapies such as chemotherapeutic agents (Elkrief et al., 2019; Gopalakrishnan et al., 2018; Helmink et al., 2019; Saus et al., 2019; Song, Chan & Sun, 2020; Viaud et al., 2013). Moreover, the metabolism of chemotherapeutic drugs can be altered by the gut or tissue microbiota, which could further determine the response of cancer cells to chemotherapy (Geller et al., 2017). In particular, Gammaproteobacteria could metabolize gemcitabine (2,2-di-fluorodeoxycytidine) into its inactive form (2,2-difluorodeoxyur-idine), suggesting that the presence of such bacteria in pancreatic adenocarcinoma (PDAC) tissue may be contributing to the PDAC resistance to gemcitabine treatment (Geller et al., 2017). Recently, bacteria have been found in the tissues of several tumor types where they plausibly play roles in shaping the chemotherapeutic drug response (Nejman et al., 2020).

Next-generation sequencing has been widely used to study the tumor microbiome, based on the 16S rRNA gene (Flemer et al., 2017; Greathouse et al., 2018; Yan et al., 2015; Zhou et al., 2019). Currently, a wide-scale bacterial 16S rRNA analysis based on multiple variable regions has been applied. This has become a standard method in bacterial taxonomic classification and identification due to its easy and rapid procedure, and the fact that it contains enough phylogenetic information (Caporaso et al., 2012; Johnson et al., 2019). Moreover, 16S rRNA analysis in combination with metabolomics can provide the estimate of microbiota functions through the changing levels of microbial and host-microbial metabolites (Langille et al., 2013). Therefore, metabolic profiling using either nuclear magnetic resonance (NMR) spectroscopy or liquid chromatography mass spectroscopy (LC-MS) can be applied to investigate the metabolic reflection of the tumor microbiota-induced drug resistance (Gong et al., 2020).

In the current study, we performed 16S rRNA sequencing of the bacteria in the tumor tissues from the CCA patients. Furthermore, an investigation of the microbial functions through metabolomic profiling was conducted. Taken together, we hypothesize that there are microbiota that can promote chemotherapeutic drug resistance, focusing on gemcitabine and cisplatin drugs for individual CCA patients. The association of the microbiota and their functions with the chemotherapeutic drug response patterns were investigated.

Materials & Methods

Patient characteristics and tissue sample collection

Thirty-six freshly frozen tissues were obtained from CCA patients who had undergone surgery at Srinagarind Hospital, Khon Kaen University during January 2017 until May 2019 and patient data have been previously described (Suksawat et al., 2019). The protocol of the specimen collection and study were approved by the Ethic Committee for Human Research, Khon Kaen University (HE601149). In addition, written informed consent was obtained from each patient prior to surgery. Fresh tumor tissues were obtained from the resection of the primary tumor and stored in Hank’s balanced salt solution (HBSS) with antibiotic (Ciproflaxin, Cefazolin and Amphotericin B) at −80 °C. In the present study, we further explored the tumor tissues based on the HDRA result from the study of Suksawat et al. (2019) which divided patients into subgroups based on chemotherapeutic response patterns. The chemotherapeutic response characteristics of CCA patients whose the intratumoral microbiota profile were analyzed using 16S rRNA sequencing and whose metabolic signature were analyzed using NMR spectroscopy are shown in Table 1.

Table 1 The characteristics of CCA patients from whom the tumor tissues were taken for the microbiome and metabolomics studies.

Variable	16S rRNA
sequencing
(n = 18)	1H NMR based
metabolomics
(n = 36)	
1,000 ug/mL gemcitabine (LDGem)			
Sensitive	6	11	
Resistant	12	25	
1,500 ug/mL gemcitabine (HDGem)			
Sensitive	4	11	
Resistant	14	25	
20 ug/mL cisplatin (LDCis)			
Sensitive	7	15	
Resistant	11	21	
25 ug/mL cisplatin (HDCis)			
Sensitive	9	16	
Resistant	9	20	
1,000 ug/mL gemcitabine plus 20 ug/mL cisplatin (Combined)			
Sensitive	13	23	
Resistant	5	13	

Histoculture drug response assay (HDRA)

Fresh tumor tissues were obtained from the resection of the primary tumor and storage in Hank’s Balanced Salt Solution (HBSS) at 4 °C. Then, the tumor tissues were minced into small pieces of approximately 9–12 mg and placed onto sponge in 24 well plates. Each well of the 24 well plates contained RPMI-1640 medium and a varying concentration of the gemcitabine and cisplatin drugs. The medium was supplemented with 20% fetal craft serum (FCS), 100 U/mL penicillin and 100 mg/mL streptomycin. After that, the tumor tissues were incubated at 37 C in 5% CO2 for 4 days. Then, 100 µL of HBSS containing 0.1 mg/mL of collagenase type I and 100 µL of MTT solution were added into each well and further incubated for 4 h. The cell viability was then measured using an MTT assay. After that, the MTT formazan products are dissolved in DMSO and subjected to absorbance measurement at 540 nm (TECAN sunrise ELISA Reader, Triad Scientific, Manasquan, NJ, USA). Finally, the percent cell growth inhibition rate was calculated as previously described (Suksawat et al., 2019). The criteria for classification sample into sensitive and resistant were previously reported (Suksawat et al., 2019). A total of thirty-six CCA tumor tissues were treated with chemotherapy in five conditions , including low dose gemcitabine (LDGem) at 1,000 ug/mL, high dose gemcitabine (HDGem) at 1,500 ug/mL, low dose cisplatin (LDCis) at 20 ug/mL, high dose cisplatin (HDCis) at 25 ug/mL and combined treatment composed of 1000 ug/mL of gemcitabine and 20 ug/mL cisplatin, and evaluated using HDRA. Tissues were then sub-classified into sensitive (S) and resistant (R) groups to a particular chemotherapeutic condition.

DNA extraction and 16s rRNA sequencing

Total DNA was isolated from approximately 50 mg fresh frozen tumor tissues following the manufacture’s protocol (QIAGEN, Hilden, Germany). For quantification of the DNA extracted a spectrophotometer (Nanodrop) was used and with 1.5% agarose gel electrophoresis for visualization. Amplification and sequencing of the V1-V2 region were conducted. Briefly, 7.5 µL of genomic DNA from tissues were amplified using the 16 rRNA gene at the variable region V1-V2 incorporating Illumina adapters and a barcode sequence amplified (forward primer:5′-TCGTCGGCAGCGTCAGATGTGTATAAGAGACAGGAGTTTGATCMTGGCTCAG-3′ and reverse primer:5′GTCTCGTGGGCTCGGAGATGTGTATAAGAGACAGGCTGCCTCCCGTAGGAGT-3′) using polymerase chain reaction (PCR) (T100TM Thermal Cycler, Bio-Rad, Hercules, CA, USA) with the specific primer using Hotstar Master Mix (QIAGEN, Hilden, Germany). The PCR cycling conditions used were: initial denaturation at 95 °C for 3 min; 25 cycles of denaturation at 95 °C for 30 s, annealing at 55 °C for 30 s, and extension at 72 °C for 30 s; and the final extension step at 72 °C for 5 min. The negative control (DNase free water) was applied in DNA extraction and 16S amplification steps. The absent band of the negative control was observed. Sequencing was performed on the Illumina MiSeq platform (Illumina®, Macrogen, Korea), with read length of 301 base pair, paired-end.

16S rRNA data processing

Following standard quality control and demultiplexing, the reads were processed using the QIIME2 (version 2021.11) pipeline (Hall & Beiko, 2018). First, paired-end reads were joined and size selected to reduce non-specific amplification. These reads were then grouped into operational taxonomic units (OTUs) based on sequence similarity using the SILVA database (version 132) (Quast et al., 2013) and classified at ≥ 99% identity of reads. Data were rarefied to the minimum library size using total sum scaling (TSS). The alpha diversity and richness of CCA tissues between resistant and sensitive groups were calculated by using Chao1 and the Shannon and Simpson diversity indices. In addition, the edgeR algorithm was applied in order to compare and classify of differential abundance between resistant and sensitive groups to chemotherapeutic treatments. To evaluate the intratumoral microbial community between resistant and sensitive groups, we used the abundance data and calculated the differential microbial composition using Bray-Curtis dissimilarity and visualized by non-metric multidimensional scaling (NMDS) on projection in MicrobiomeAnalyst (Chong et al., 2020; Dhariwal et al., 2017).

Metabolite extraction and metabolomics analysis

Approximately 100 mg of each fresh frozen tumor tissue was used for metabolite extraction. The tumor tissues were then homogenized using a Dounce homogenizer and extracted by adding 400 µL of methanol and 85 µL of HPLC grade water, followed vortex mixing. Then, 200 µL of chloroform and 200 µL of HPLC grade water were added followed by vortex mixed. Next, the tissue extracted solutions are transferred into 15 mL tubes and sonicated 3 times using the following parameters: sonicate on 30 s and sonicate off 10 s at amplitude 40% and temperature of 4 °C. After that, the 15 mL tubes were subjected to centrifugation at 1,000 g at 4 °C for 15 min. The aqueous phase was subjected to nuclear magnetic resonance (NMR) spectroscopy or global profiling analysis. The NMR spectra data acquisition from NMR used peak alignment, normalization with probablistic quotient normalization and scaling using matrix laboratory software (MATLAB) (MathWorks Inc., US). The significant metabolites were identified using statistical total correlation spectroscopy (STOCSY), human metabolome database (HMDB) (Wishart et al., 2018; Wishart et al., 2013; Wishart et al., 2009; Wishart et al., 2007) and the Chenomx NMR suite (Chenomx Inc., Canada). The pairwise comparison of the log2 transformed data of metabolites between the resistant and sensitive groups was conducted with a paired non-parametric test (Mann–Whitney U test) and adjusted p value was calculated with a Benjamini–Hochberg procedure. The data was illustrated using Graph Pad prism 5 (GraphPad Software, Inc., CA, US). The network analysis was performed using Metscape (Gao et al., 2010) for visualizing metabolic pathways.

Correlation analysis

The correlation analysis was performed with Spearman’s correlation coefficient at the genus level and metabolites using the M2IA pipeline (Ni et al., 2020) for the integrated microbiome and metabolome dataset.

Results

Difference of intratumoral microbiota composition between resistant and sensitive group of chemotherapeutic treatment in cholangiocarcinoma patients

Out of 36 tumor tissues, amplification for V1–V2 regions was successful for 18 samples. These samples were sequenced and a total read of 3,504,888 were acquired for microbial profiling. Following quality trimming and merging of overlapping paired-end reads, total read counts of 540,202 counts were retained from 18 samples, average counts per sample 30,011 counts. These reads could be assigned into a total of 890 bacterial OTUs. Overall, the intratumoral microbiome profile revealed a common pattern with the phyla Proteobacteria, Actinobacteria and Firmicutes dominating in both the resistant and sensitive groups in all conditions of chemotherapeutic treatment (Figs. 1A and 1D). The top three most abundant classes were Gammaproteobacteria, Actinobacteria and Alphaproteobacteria (Figs. 1B and 1E). The intratumoral microbiome profile in genera were shown in Figs. 1C and 1F. We then compared the alpha diversity between the resistant and sensitive groups. The Shannon and Simpson indexes revealed that tumor tissues treated with LDGem and HDGem had significant differences in microbial diversity between the resistant and sensitive groups. In contrast, Chao1 index demonstrated no difference in species richness between the resistant and sensitive groups (Fig. 2). A comparison of taxonomic profiles at the phylum level revealed that LDGem resistant group, HDGem resistant group, LDCis resistant group and HDCis resistant group showed higher abundance of Proteobacteria. A comparison of the taxonomic profiles at the class level demonstrated that tumor tissues which were resistant to LDGem, HDGem and LDCis exhibited higher abundances of Gammaproteobacteria, whereas the abundances of Actinobacteria was found to be lower in LDGem resistant group and HDGem resistant group (Fig. 3).

Figure 1 Taxonomic composition of the intratumoral bacteria in cholangiocarcinoma tissues.

Taxonomic composition of the intratumoral bacteria in cholangiocarcinoma tissues. Stacked bar plot of taxonomic relative abundance (A) phylum level (B) class level (C) genus level. The heatmap and hierarchical clustering represent the relative abundance of intratumoral microbiota, which each row demonstrated the taxonomic unit and each column represent the sample at (D) phylum level (E) class level (F) genus level. The resistant and sensitive groups were color-coded in red and blue, respectively, and indicated on top of heatmap. The heatmap color spectrum (blue to darked) represents the relative abundance of each taxon. The clustering was constructed based on Euclidean distance.

Figure 2 The microbial alteration in cholangiocarcinoma based on chemotherapeutic treatments.

The alpha diversity index of the relative abundance from cholangiocarcinoma tissues was analysed by the Kruskal–Wallis (pairwise) test. An adjusted P-value less than 0.05 was considered as statistically significant.

Figure 3 Intratumoral bacteria between the resistant and sensitive groups at the phylum and class levels.

The significant difference of log2 fold differential abundance was analysed by edgeR algorithm of microbiome analyst based on adjusted P values.

To explore whether the intratumoral microbial composition of CCA patients was different between the resistant and sensitive groups, non-metric multidimensional scaling (NMDS) was performed. NMDS is based on Euclidean distance and can reveal a shift of centroid (indicated by arcs) and variation in the microbiota community profiles of each chemotherapeutic drug treatment condition (circled area). The NMDS analysis at the class level demonstrated the overlap of the circle areas in each plot between the sensitive and resistant groups, showing some similar bacterial communities between the sensitive and resistant groups in all chemotherapeutic treatment conditions except, the resistant group of HDCis showed the smallest variance in the bacterial community (Fig. 4).

Figure 4 The non-metric multidimensional scaling (NMDS) plot based on Euclidean distance (β-diversity) at class level. (A) LDGen (B) HDGem (C) LDCis (D) HDCis (E) combined.

Metabolic alteration associated with chemotherapeutic responses

1H NMR metabolic signatures from the CCA tissues are represented in Table 2. The metabolic differences between resistant and sensitive groups of CCA patients can be distinguished on univariate analysis (Mann–Whitney U test) using a log2 transformation of maximum intensity. Significantly higher levels of acetylcholine, adenine, carnitine and inosine were observed in the LDGem resistant group. For the LDCis treatment, the levels of acetylcholine, alpha-D-glucose and carnitine were significantly increased in the resistant group compared to the sensitive group (Fig. 5). Towards the understanding of host-bacterial altered metabolic profiles, we performed metabolic pathway analysis executed on Metscape using KEGG (Kyoto Encyclopedia of Genes and Genomes) pathways, to investigate the most relevant pathways triggered by the chemotherapeutic response conditions. In addition to the upregulated acetylcholine metabolism and carnitine metabolism in both LDGem and LDCis groups, LDGem group exhibited the enhanced inosine and adenine metabolism and glucose metabolism (Fig. 6). Therefore, adenine and inosine involved in nucleotide metabolism also promote cancer cell proliferation (Newman & Maddocks, 2017). In addition, carnitine indicated cancer development and progression (Kawai et al., 2017). In term of glucose, glucose serve as inducer of progression of CCA (Saengboonmee et al., 2016). Furthermore, acetylcholine can promote cancer stem cell proliferation (Nguyen et al., 2018).

Table 2 List of all metabolites that were found in NMR spectra of CCA tumor samples.

NO.	1H chemical shift	Metabolites	
1.	0.942 (t)a, 0.994 (d)a, 1.039 (d)a, 1.261 (m)b, 1.478(m)a, 1.963 (m)b, 3.615 (d)a	Isoleucine	
2.	0.955 (t)a, 1.671 (m)b, 3.73 (m)a	Leucine	
3.	0.987 (d)a, 1.038 (d)a, 2.247 (m)a, 3.614 (d)a	Valine	
4.	1.327 (d)a, 4.103 (q)a	Lactate	
5.	1.478 (d)a, 3.754 (q)a	Alanine	
6.	1.923 (s) a	Acetate	
7.	2.105 (m)a, 2.358 (dt)a, 3.763 (t)a	Glutamate	
8.	2.113 (m)a, 2.635 (t)b, 3.832 (dd)a	Methionine	
9.	2.340 (m)a, 2.077 (m)a, 3.329 (dt)a, 3.401 (m)a, 4.120 (dd)b	Proline	
10.	2.408 (s) a	Succinate	
11.	2.520 (d)a ,2.664 (d)a	Citrate	
12.	3.040 (s)a, 3.935 (s)a	Creatine	
13.	3.188 (s)a, 3.514 (dd)a, 4.063 (m)a	Choline	
14.	2.163 (s)a, 3.230 (s)a, 3.74(t)a, 4.56 (m)b	Acetylcholine	
15.	2.421(s)a, 3.215(s)b, 3.231 (s)a, 3.414(s)a, 4.555(s)b	Carnitine	
16.	3.258 (t)a, 3.414 (t)a	Taurine	
17.	3.033 (dd)a, 3.280(dd)a, 3.289(dd)a, 3.304 (dd)a, 3.554 (dd)a, 3.720(dd)a, 4.103 (dd)a	Cysteate	
18.	2.730 (s)b, 3.614 (s)a	Sarcosine	
19.	2.142 (m)a, 2.446 (m)a, 3.754 (t)a	Glutamine	
20.	3.029 (s)b, 3.934 (s)a	Phosphocreatine	
21.	2.827 (d)a, 2.853 (s)a, 2.874(s)a, 2.930 (d)b, 2.960 (d)b, 3.973 (dd)a	Asparagine	
22.	3.239 (dd)a, 3.396 (m)a, 3.456 (m)a, 3.532 (dd)a, 3.720 (m)a, 3.820 (m)a, 4.648 (d)b, 5.240 (d)a	Alpha-glucose	
23.	6.524 (s) a	Fumarate	
24.	3.037 (d)a, 3.062 (d)a, 3.205 (dd)a, 3.935 (dd)a, 6.914 (d)a, 7.191 (d)a	Tyrosine	
25.	5.803 (d)a, 7.542 (d)a	Uracil	
26.	2.470(s)b, 7.688 (s)a	Pyridoxine	
27.	3.140(dd)a, 3.247(dd)a, 3.972 (dd)a, 7.900 (s)b, 7.08 (s)b, 7.841 (s)a	Histidine	
28.	2.827(m)a, 3.140 (m)a, 3.515(s)a, 7.130(m)b, 7.840 (m)a	Thyroxine	
29.	3.487(s)a, 3.783(d)a, 3.917(d)a, 4.108(dd)b, 4.620(td)b, 6.070 (d)a, 6.097(d)a, 9.580(d)b	Uridine	
30.	1.893(m)a, 2.340(m)a, 2.900(m)a,3.003(dd)a, 3.188(dd)a,4.480(m)a, 7.901 (s)a	Homocarnosine	
31.	8.245 (s) a	Adenine	
32.	3.823(dd)a,3.900(dd)a,4.259(dd)a, 4.420(dd)b, 6.098 (d)a, 8.187(s)a, 8.351 (s)a	Inosine	
33.	8.461 (s) a	Formate	
Notes.

s, Singlet; d, Doublet; dd, Doublet of doublet; t, Triplet; q, Quartet; m, Multiplet.

a Resonances that were identified in both STOCSY and HMDB.

b Resonances that were identified only in HMDB.

Bold text represents chemical shift that were selected to analysis.

Figure 5 Significantly changed metabolites in LDGem and LDCis from tumor tissues of CCA patients.

The blue color shows sensitive group and red color shows resistant group. An asterisk (*) indicates statistically significant (adjusted P value < 0.05).

Figure 6 The metabolic pathway constructed by Metscape.

(A) the metabolic network of LDGem resistance group (B) the metabolic network of LDCis resistance group. The red box represents significantly increased metabolites in resistance group (adjusted P value < 0.05).

Correlation of metabolic profile and intratumoral microbiota composition

To examine the overall correlation between tissue microbial and metabolic profiles and to identify the accountable microbiota and metabolite(s), we performed a Spearman-rank correlation analysis between the genus-level relative abundances of tissues microbiota and the log2 transformed relative concentrations of metabolites. In LDGem, Deinococcus was negatively correlated with homocarnosine and L-methionine, and Escherichia-Shigella was negatively correlated with homocarnosine (Fig. 7A). In HDGem, Deinococcus and Pseudomonas were negatively correlated with acetic acid and L-methionine; Atopostipes and Paracoccus were negatively correlated with acetic acid; and Streptococcus was negatively correlated with L-methionine (Fig. 7B). Finally, in HDCis, Cutibacterium was found to be positively correlated with L-leucine and L-isoleucine (Fig. 7C). There was no observable correlation between microbiome and metabolites in the LDCis and combined groups.

Figure 7 Spearman-rank correlation analysis between the genera of the intratumoral microbiome and metabolites by chemotherapeutic treatments.

(A) LDGem (B) HDGem (C) HDCis. An asterisk (*) indicates significant correlation. The color is based on the Spearman-rank correlation coefficient between significant changes for genera and metabolites; blue represents a significantly negative correlation (adjusted P < 0.05), red a significantly positive correlation (adjusted P < 0.05).

Discussion

Host metabolism has been known to interact with the gut microbiota, which can, in turn, affect host disease status (Elia & Haigis, 2021; Zhao, 2013). In the present study, we performed metabolome analysis in 36 tumor tissues and microbiome analysis in 18 tumor tissues of CCA patients. We elucidated the microbial community using 16S rRNA sequencing and metabolic profiles using NMR-based metabolomics. The exploration of intratumoral microbiome of CCA tumor with 16S rRNA sequencing allows us to compare resistant and sensitive groups of chemotherapeutic treatment condition. Based on our results using 16S rRNA sequencing, a significant difference occurred in α-diversity and β-diversity in gemcitabine treatment responses comparing resistant and sensitive subgroups. Interestingly, the intratumoral microbiota shift was found in the CCA tissues which resisted the chemotherapeutic drug treatment. Our findings are consistent with the previous study in which the microbiota dysbiosis was correlated with CCA progression and pathogenesis (Saab et al., 2021). Microbial community at the phylum level demonstrated a common pattern of microbiota composition between the resistant and sensitive groups of chemotherapeutics treatment. However, the relative abundance of the class Gammaproteobacteria was significantly higher in the resistant group to gemcitabine treatment. Our results conform with a previous study in pancreatic ductal adenocarcinoma (PDAC) (Geller et al., 2017). The Gammaproteobacteria, the most common bacteria found in gemcitabine resistant PDAC tissues, can express cytidine deaminase (CDD) enzyme in its long form (CDDL) which can metabolize the active form of gemcitabine into the inactive form (Choy et al., 2018). The present work was limited by the low amount of bacterial DNA extracted from tumor tissues, resulting in some difficulties during the amplification, which may affect the power in finding more candidate phyla from the microbial profiles. Moreover, a future study in larger cohorts will help further validate the sensitivity and specificity of biomarkers based on microbial composition.

We further investigated the metabolic differences and their biological relevance in the chemotherapeutic drug response pattern. In regards with the NMR-based metabolomics, the levels of acetylcholine, adenine, carnitine and inosine were increased with gemcitabine resistance, while the levels of acetylcholine, alpha-D-glucose and carnitine were increased with cisplatin resistance. Expectedly, we found significantly increased amino acid levels in the resistant group of gemcitabine and cisplatin treatment, that is consistent with a previous study showing the elevated amino acid levels in a resistant group of both chemotherapeutic drugs (Ciccarone et al., 2017). Moreover, we found a significantly higher levels of nucleotides in CCA that were resistant to gemcitabine. The previous study indicated that nucleotide metabolites also promote cancer cell proliferation (Newman & Maddocks, 2017). We also found a significantly higher glucose level in the cisplatin resistant group, which is consistent with previous studies that demonstrated lung cancer patients who are resistant to platinum-based combination chemotherapy shown elevated of glucose level was found in serum and increased of glucose level in CCA patients associated with progression of CCA in an in vitro study (Saengboonmee et al., 2016; Xu et al., 2017). Acetylcholine may also serve as an inducer of cancer stem cell proliferation (Nguyen et al., 2018). Even though the evidence of carnitine in chemotherapy response has not been widely studied, a previous study has shown that patients responding to cisplatin therapy had lower levels of carnitine in gastric cancer patients and it has been defined as an oncometabolite that is involved in cancer development and progression (Kawai et al., 2017). In conclusion, the metabolic profiles could reflect the drug response patterns of CCA patients’ tissues and may serve as predictive biomarkers for chemotherapeutic drug response.

Based on an integration analysis between intratumoral microbiota and metabolites data related to the drug response pattern, Streptococcus and Deinococcus were negatively correlated with L-methionine. Previous work showed that Streptococcus could take up L-methionine through ABC transport lipoprotein, which reflects the decreased level of L-methionine (Basavanna et al., 2013). We also found that Cutibacterium was positively correlated with L-isoleucine and L-leucine in the cisplatin treatment group. Bacteria in the Cutibacterium phyla (formerly Propionibacterium) have been reported to be able to trigger the catabolism of leucine and isoleucine metabolic pathway from substrates available in the colon environment (Saraoui et al., 2013). Escherichia-Shigella was negatively correlated with homocarnosine. Presently, there is no study, to our knowledge, that demonstrates the interaction between homocarnosine and Escherichia-Shigella. Furthermore, Pseudomonas, Atopostipes, Paracoccus and Deinococcus were negative correlated with acetic acid in the high dose gemcitabine treatment group, reflecting the alteration of intestinal microbiota as evident by a previous study in colorectal cancer patients (Yusof et al., 2018). However, there is no report on the association of acetic acid, which could induce microbiota composition change in cholangiocarcinoma. The relationship between the response pattern to chemotherapy from HDRA and clinical drug response of CCA patients in a prospective manner requires further study. Moreover, validation studies will need to be performed in larger cohorts (pre- and post-treatment) prior to the actual clinical use. Ultimately, such data will serve the development of effective ways and less invasive tools to be eventually applied in the clinical application.

Conclusions

An integration of the omics studies potentially provides an understanding of the alteration of host metabolic changes and microbiota composition shifts during disease progression. The present study provides an insight into the correlation between the metabolic changes and microbial alterations in the CCA tissues and its potential effects on the chemotherapeutic treatments. The disruption of the intratumoral microbiome, metabolites, functional analysis and the clinical chemotherapy outcomes could be further validated in a larger cohort to improve the stratified treatment regimen for individual patients. Moreover, the drug resistance biomarker detection of biological fluids including plasma, serum, urine, bile fluid needs to be explored in order to find a quick, effective and less invasive strategy to be eventually applied in the clinical application.

The authors express gratitude to Professor Trevor N. Petney for editing the MS via the Publication Clinic KKU, Thailand.

Additional Information and Declarations

Competing Interests

Author Contributions

Human Ethics

DNA Deposition

Data Availability

The authors declare there are no competing interests.

Sirinya Sitthirak performed the experiments, analyzed the data, prepared figures and/or tables, authored or reviewed drafts of the article, and approved the final draft.

Manida Suksawat analyzed the data, authored or reviewed drafts of the article, and approved the final draft.

Jutarop Phetcharaburanin conceived and designed the experiments, performed the experiments, analyzed the data, prepared figures and/or tables, authored or reviewed drafts of the article, and approved the final draft.

Arporn Wangwiwatsin performed the experiments, analyzed the data, prepared figures and/or tables, authored or reviewed drafts of the article, and approved the final draft.

Poramate Klanrit performed the experiments, analyzed the data, authored or reviewed drafts of the article, and approved the final draft.

Nisana Namwat performed the experiments, analyzed the data, authored or reviewed drafts of the article, and approved the final draft.

Narong Khuntikeo analyzed the data, authored or reviewed drafts of the article, specimen enqury and Visualization, and approved the final draft.

Attapol Titapun analyzed the data, authored or reviewed drafts of the article, specimen enqury and Visualization, and approved the final draft.

Apiwat Jarearnrat analyzed the data, authored or reviewed drafts of the article, specimen enqury and Visualization, and approved the final draft.

Sakkarn Sangkhamanon analyzed the data, prepared figures and/or tables, authored or reviewed drafts of the article, specimen enqury and Visualization, and approved the final draft.

Watcharin Loilome conceived and designed the experiments, performed the experiments, analyzed the data, prepared figures and/or tables, authored or reviewed drafts of the article, and approved the final draft.

The following information was supplied relating to ethical approvals (i.e., approving body and any reference numbers):

The protocol of the specimen collection and study were approved by the Ethic Committee for Human Research, Khon Kaen University (HE601149).

The following information was supplied regarding the deposition of DNA sequences:

The 16S rRNA sequencing data, adapters trimmed, are available at the Sequence Read Archive (SRA): PRJEB47824.

They are also available at the European Nucleotide Archive: ERP132128.

The following information was supplied regarding data availability:

Sequencing data is available at NCBI: PRJEB47824.

Metabolomic data is available at Open Science Framework (OSF):

SIRINYA. “Chemotherapeutic Resistant Cholangiocarcinoma Displayed Distinct Intratumoral Microbial Composition and Metabolic Profiles.” OSF, December 5, 2021. https://osf.io/6uxbr/.

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
