# Peer review of "Chemotherapeutic resistant cholangiocarcinoma displayed distinct intratumoral microbial composition and metabolic profiles"

_PeerJ, doi:10.7717/peerj.13876_

## Round 0.1 · original submission · Major Revisions

Overall your paper was well-received but both reviewers felt that changes were needed.

Reviewer 1 ·

Basic reporting

The manuscript entitled “Chemotherapeutic resistant cholangiocarcinoma displayed distinct intratumoral microbial composition and metabolic profiles (#72279)” by Sitthirak and colleagues is an interesting and timely report that will be of interest to PeerJ readership. The manuscript generally conforms to PeerJ standards, although some editing for punctuation, grammar and clarity is needed.
The report describes a set of experiments designed to evaluate potential effects of intratumoral bacteria on resistance to two frontline chemotherapeutic agents (gemcitabine and cisplatin) used to treat CCA. Although prior work has demonstrated GI and tumor-associated microbes can alter the efficacy of immune and chemotherapeutic agents in other types of cancer, it is currently unknown if bacteria play a role in therapy response in CCA. To this end, the study presented in this paper is novel.
A concise abstract of the study’s methods, results, and conclusion is clearly given.
The Introduction and background section gives appropriate context for the study and cites the major relevant literature in the field.
Appropriate bioinformatic and statistical analyses were performed and presented with sufficient detail to replicate experiments. However, the paper would benefit from a more detailed description of tissue sample collection, including i) time period of sample collection; ii) patient data such as age, sex, and tumor stage.
Also, I suggest merging the entire “Patient Characteristics” paragraph (Lines 216-222) within the “Tissue sample collection” subheading and re-title the subsection. “Patient characteristics and tissue sample collection”
Line 193-194, the authors state, “This was repeated if there was still no clear separation. Then, samples will be separated into the aqueous and the lipophilic phases.” This detail is unnecessary, suggest these sentences be deleted.

Experimental design

There are no major concerns relative to the experimental design of the study.
The manuscript presents original findings from the metagenomic and metabolomic analyses of CCA clinical samples and addresses significant questions regarding a potential role for intratumoral bacteria in the efficacy of chemotherapy.
The data generated from this study provides only correlative support of the hypothesis that there are microbiota that can promote resistance to the drugs gemcitabine and cisplatin in individual CCA patients. The hypothesis is not mechanistically tested, which limits the significance of these findings. In addition, the manuscript would be strengthened by presentation of underlying data on the relative abundances of microbial genera or defined OTUs (only phyla and class level abundances are presented in Figs. 2,3 & 4). This would clarify the analysis and discussion of genus-level correlations, e.g. Line 278-281 the authors state, “we performed a Spearman-rank correlation analysis between the genus-level relative abundances of tissues microbiota and the log2 transformed relative concentrations of metabolites.”

Line 371-372, the authors state, “The present study provides an insight into the association between the human metabolic changes and microbial alterations in the CCA tissues...” However, the NMR and KEGG analysis does not distinguish between host and microbial-derived metabolic activity. Experiments that show the microbial contribution to tumor metabolite pools are needed to confirm these conclusion statements.

A major limitation is the retrospective nature of the study, so no measurements or comparisons are provided of microbe and metabolite levels prior to treatment, making insights and interpretations about shifts in levels impossible. E.g. Line 295-297, the authors state, “Therefore, an integration of the omics studies potentially provides an understanding of the alteration of host metabolic changes and microbiota composition shifts during disease progression.”

Validity of the findings

The manuscript presents a well developed idea that combining metagenomic and metabolomic data from tumor tissue has potential for identifying biomarkers of chemotherapeutic response.

The conclusions identify important questions and gaps in knowledge for the field, and identify specific areas of tumor-microbe interactions that warrant further investigation for clinical development.

Overall, the correlations of metabolic alteration associated with chemotherapeutic responses is strong, however the integration of metagenomic data with metabolite pools is less convincing. Additional data on abundances of microbial genera present or expression levels of KEGG-identified metabolic pathways in tumor-associated microbes would strengthen the paper considerably.

Line 265-267, the authors state, “Significantly higher levels of acetylcholine, adenine, carnitine and inosine were observed in the LDGem resistant group.” Some discussion of the biological relevance of the measured levels of specific metabolites should be included here (not only in the discussion section). What is indicated by higher levels of acetylcholine, adenine, carnitine and inosine – and how does this relate to drug resistance/sensitivity?

Line 278-281 the authors state, “we performed a Spearman-rank correlation analysis between the genus-level relative abundances of tissues microbiota and the log2 transformed relative concentrations of metabolites.” Line 305-307, “Interestingly, the intratumoral microbiota shift was found in the CCA tissues which resisted the chemotherapeutic drug treatment.”
Some underlying data on the relative abundances of microbial genera is not presented (only phyla and class level abundances are presented in Figs. 2,3 & 4), despite analysis and discussion of genus-level correlations.

Additional comments

General Comments:
In general the science in the report is solid and would contribute to the literature on microbial effects on drug toxicity and efficacy profiles.

The manuscript needs some editing to improve its clarity, particularly in the discussion section (see comments) and to provide more context for how the experimental results are a significant advance to the field of microbial effects on chemotherapy.

Specific comments:
Line 36-37, The authors state, " levels of acetylcholine, alpha-D-glucose and carnitine increased in the low dose cisplatin-resistant group.” It is not clear that these results are consistent with results cited (Suksawat et al., 2019) that showed TCA intermediates alpha-D-glucose and ethanol may serve as predictive biomarkers for gemcitabine and cisplatin sensitivity in the tumor tissue of CCA. Were levels there also increased?

Line 193-194 “This was repeated if there was still no clear separation. Then, samples will be separated into the aqueous and the lipophilic phases.” These sentences can be deleted.
Line 228, The authors state, “These samples were proceeded to sequencing…” Change to, “These samples were sequenced…”

Lines 292-293, The authors state, “ Host metabolism has been known to interact with the gut microbiota, which can, in turn, affect host disease status (Zhao, 2013).” Also consider the work from Elia I, Haigis MC. Nat Metab. 2021 Jan 3(1):21-32. doi: 10.1038/s42255-020-00317-z. Epub 2021 Jan 4. PMID: 33398194.

Reviewer 2 ·

Basic reporting

Major concern:
1. How did the results of histoculture drug response assay (HDRA) correlate with actual therapeutic resistance and/or survival of the patient cohort in this study?
2. What hypothesis can be derived from the results for future testing?

Technical questions:
1. Please use receiver operating characteristic (ROC) curve to evaluate the classification capability of microbial taxonomic composition on drug resistance in vitro.
2. Is there any metabolite signature correlated with specific bacterial taxonomy?

Experimental design

This is a correlation study. No specific issue in the experimental design was found.

Validity of the findings

This is a correlation study. The results need to provide suggestions for future study.

Additional comments

N/A

---

## Round 0.2 · Minor Revisions

Thank you for submitting to PeerJ. We are pleased to publish the article upon minor revisions and we hope that you will consider the Journal for future submissions.

Reviewer 1 ·

Basic reporting

The resubmitted manuscript entitled “Chemotherapeutic resistant cholangiocarcinoma displayed distinct intratumoral microbial composition and metabolic profiles (#72279)” by Sitthirak and colleagues is an interesting and timely report that will be of interest to PeerJ readership. The manuscript generally conforms to PeerJ standards, and has been significantly improved in clarity and presentation of the novel findings.
The report describes a set of experiments designed to evaluate the association and potential effects of intratumoral bacteria on resistance to two frontline chemotherapeutic agents (gemcitabine and cisplatin) used to treat CCA. Although prior work has demonstrated GI and tumor-associated microbes can alter the efficacy of immune and chemotherapeutic agents in other types of cancer, it is currently unknown if bacteria play a role in therapy response in CCA. To this end, the study presented in this paper is novel.
A concise abstract of the study’s methods, results, and conclusion is clearly given.
The Introduction and background section gives appropriate context for the study and cites the major relevant literature in the field.
Appropriate bioinformatic and statistical analyses were performed and presented with sufficient detail to replicate experiments.

Experimental design

There are no major concerns relative to the experimental design of this correlation study.
The manuscript presents original findings from the metagenomic and metabolomic analyses of CCA clinical samples and addresses significant questions regarding a potential role for intratumoral bacteria in the efficacy of chemotherapy.
Line 122: Replace “As the present study…” with “In the present study…”

Validity of the findings

The manuscript presents a well developed idea that combining metagenomic and metabolomic data from tumor tissue has potential for identifying biomarkers of chemotherapeutic response.
Line 336: “patients who resistant to platinum-based…” should read, “patients who are resistant to platinum-based…”
Line 341-342: “carnitine in chemotherapy response has not been widely studied, In a previous study, it was shown that when patients…” should read, “carnitine in chemotherapy response has not been widely studied, a previous study has shown that patients responding to cisplatin therapy had lower levels of carnitine …”
Line 361: “ negative correlated” should read, “negatively correlated”

The conclusions identify important questions and gaps in knowledge for the field, and identify specific areas of tumor-microbe interactions that warrant further investigation for clinical development. Since this is a correlation study, it should identify future areas of research focus.
Suggest adding to Line 372: “The relationship between the response pattern to chemotherapy from HDRA and clinical drug response of CCA patients in a prospective manner requires further study.”

Additional comments

In general the science in the report is solid and would contribute to the literature on microbial effects on drug toxicity and efficacy profiles.
Some minor editing to improve clarity remains, (see comments).

Reviewer 2 ·

Basic reporting

The report is written well and the rationale is clear.

Experimental design

The experimental design is good.

Validity of the findings

This is a descriptive study. The ROC curve analysis should be discussed.

Additional comments

The major concern of this study, together with the previous two from the same group (PLoS One, 14(9), e0222140, 2019. and Front Public Health, 10, 766023, 2022.) is the lack of direct relation between patient responses and predictors. The logic flow of these studies goes like this:

patient response to gemcitabine => known predictors (e.g. expression of DCK, hENT-1, and RRM1) => histoculture drug response assay (HDRA) => correlation in microbiome

First, the predictive power of HDRA on patient response to chemo drug is never tested directly. Second, even the correlation between HDRA and known predictors is weak at best. Therefore, this is a descriptive study and the clinical relevance needs further investigation.

However, it doesn't mean the study has no practical value. If the authors can follow the patients' responses to chemotherapy and run a retrospective analysis, it will be very helpful to make HDRA a very useful clinical assay. Moreover, if the authors can compare the response of xenograft in mice to HDRA, and find they are well correlated, HDRA can replace many mouse studies.

Finally, the author should consider RNA sequencing for the tissues used in HDRA, and compare the gene expression with published RNA seq data. Many of RNA seq data are from patients receiving chemotherapy. This could be an easier way of validating HDRA.

The authors should consider discussing these points in their manuscript.

---

## Round 0.3 · accepted · Accept

Thank you for addressing all the concerns raised by the reviewers. I congratulate you on the acceptance of your manuscript for publication.